# xTED: Cross-Domain Adaptation via Diffusion-Based Trajectory Editing

## Abstract

Reusing pre-collected data from different domains is an appealing solution for decision-making tasks that have insufficient data in the target domain but are relatively abundant in other related domains. Existing cross-domain policy transfer methods mostly aim at learning domain correspondences or corrections to facilitate policy learning, such as learning domain/task-specific discriminators, representations, or policies. This design philosophy often results in heavy model architectures or task/domain-specific modeling, lacking flexibility. This reality makes us wonder: can we directly bridge the domain gaps universally at the data level, instead of relying on complex downstream cross-domain policy transfer models? In this study, we propose the **Cross**-Domain **T**rajectory **ED**iting (**xTED**) framework that employs a specially designed diffusion model for cross-domain trajectory adaptation. Our proposed model architecture effectively captures the intricate dependencies among states, actions, and rewards, as well as the dynamics patterns within target data. By utilizing the pre-trained diffusion as a prior, source domain trajectories can be transformed to match with target domain properties while preserving original semantic information. This process implicitly corrects underlying domain gaps, enhancing state realism and dynamics reliability in the source data, and allowing flexible incorporation with various downstream policy learning methods. Despite its simplicity, xTED demonstrates superior performance in extensive simulation and real-robot experiments.

## 1 Introduction

Policies learned on insufficient data from a single domain easily end up with undesirable performance, especially in real systems (Guiochet et al., 2017; Zhan et al., 2022) where performing reinforcement/imitation learning (RL/IL) in the real world (i.e. target domain) yields laborious and costly data collection, notorious reset, and reward specification issues (Zhu et al., 2020). Faced with restricted data acquisition in reality, people resort to incorporating data generated with simulation or pre-collected from other domains (i.e. source domains) into policy learning for greater synergy (Niu et al., 2022; 2023; Open X-Embodiment et al., 2023). However, source domains invariably bear multiple domain gaps (Niu et al., 2024), such as appearance (Tobin et al., 2017) and dynamics gaps (Peng et al., 2018), as well as morphology gaps for embodied agents (Gupta et al., 2021). These domain discrepancies would reduce the availability of source data since directly augmenting target data with unprocessed source data sometimes leads to negative effects on policy learning.

Existing cross-domain policy learning methods tend to design domain-specific policy transfer models equipped with domain correspondences (Zhang et al., 2020), corrections (Eysenbach et al., 2021), or discriminations (Stadie et al., 2016; Sharma et al., 2019) that either rely on task-specific architecture designs (e.g. special encoders (Mueller et al., 2018; Wang et al., 2022a) and domain-specific regularizations (Desai et al., 2020; Eysenbach et al., 2021; Niu et al., 2022; Xue et al., 2023)), or are only applicable to specific domain of data (e.g. adaptable vision encoder design only applicable to image inputs (Rao et al., 2020). Additionally, the domain-specific discriminators or mappings (Stadie et al., 2016; Liu et al., 2018) sometimes require tedious efforts to refresh for accomodating multiple source domains, which substantially hinders data reutilization efficiency and efficacy. These limitations highlight the need for a more generic and flexible approach that addresses domain gaps more directly.

Above inadequately addressed issues make us wonder: *Rather than relying on sophisticated and domain-specific policy transfer models, is it possible to bridge domain gaps at the data level?* If source trajectories can be transformed to minimize domain gaps and seamlessly utilized for policy

Figure 1: The connections and distinctions between image editing and trajectory editing.

learning, any policy learning method can be flexibly chosen with primary focus on task-relevant considerations, without the burden of cross-domain design complexities. To achieve this goal, we tend to correct the biased domain modeling within trajectories based on target data while preserving valuable and primitive information from source domains. With these pursuits, a natural analogy can be drawn that domain transfer at the trajectory level parallels style transfer in image editing (Meng et al., 2022), where only the visual style and aesthetic elements of an image are transformed without altering its narrative content, such as converting a stroke-painted castle into a photorealistic one. However, as depicted in Fig. 1, source-domain trajectories face not only biased observation styles but also altered viewpoints and inaccurate transition dynamics over time. Moreover, trajectories are composed of fundamentally heterogeneous elements—such as observations, actions, and rewards—that, if naïvely treated as equivalent pixels in an image, would obscure the recognition of their respective inherence and delicate interactive dependencies. Thus, despite sharing a similar high-level philosophy, this editing solution is not directly applicable to cross-domain policy transfer due to the intrinsic heterogeneity and dependencies within trajectories and the complex domain discrepancies.

To address these distinctions, we propose the **Cross**-Domain **T**rajectory **ED**iting (**xTED**) paradigm with a novel diffusion model architecture tailored for decision-making data, which effectively captures the target trajectory distribution as a prior. The architecture illustrated in Fig. 2 employs a separate encoding-decoding approach to recognize states, actions, and rewards, each possessing fundamentally different physical meanings and unique internal consistency patterns. To effectively capture the dynamics patterns within trajectories, we implement dependency structure modeling mechanisms that capture the interactive relationships among states, actions, and rewards. With the assistance of above designs, we successfully extend the philosophy in diffusion-based image editing (Huang et al., 2024) to decision-making data, equipping xTED pipeline with three simple steps: (1) train diffusion model on the target data; (2) distort source data with noises and then denoise them with the pre-trained diffusion model; (3) incorporate edited source data into target data for policy learning with any algorithm at will. Through extensive experiments, we show that incorporating source data edited with xTED consistently yields performance improvements over training solely on target data while directly adding unprocessed source data often results in significant performance degradation, particularly in real-robot manipulation tasks.

## 2 RELATED WORK

### 2.1 CROSS-DOMAIN POLICY ADAPTATION

Addressing the challenges posed by domain gaps has long been established as an essential task for the prolonged development of data-hungry policy learning methods and models (Niu et al., 2024). The most straightforward approach is to construct direct mappings for state and action space between source and target domains (Liu et al., 2018; Kim et al., 2020; Zhang et al., 2020; Raychaudhuri et al., 2021; Wang et al., 2022b). An alternative is to learn domain-agnostic task-relevant embedding with mutual information criterion (Franzmeyer et al., 2022) and explicit domain discrimination (Stadie et al., 2016; Sharma et al., 2019), which can be achieved with temporal contrastive learning methods (Sermanet et al., 2018; Dwibedi et al., 2019; Yang et al., 2023; Choi et al., 2023; Li et al., 2024) to enable representation highly dependent on task progress while neglecting domain-variant information. Other than representation learning, it is also preferable to directly regularize the policy learning process in situations with mild observation gaps, by reward augmentation (Eysenbach et al., 2021; Liu et al., 2022; Xue et al., 2023) and reweighting the value update (Niu et al., 2022; 2023; Xu et al., 2023a). The complicated domain/task-specific designs hinder convenient reuse and fine-tuning

of those models to accommodate data from multiple source domains available at different stages of training (Niu et al., 2024). In contrast, xTED introduces a generic and flexible approach that handles domain gaps at the data level, avoiding any task-specific design and domain-specific fine-tuning while applicable to multiple source domains of data and freeing the choices of any upstream observation encoder and downstream policy learning method.

## 2.2 DIFFUSION MODELS FOR DECISION MAKING

Diffusion models have been applied across various decision-making settings, including generating multi-modal policies (Chi et al., 2023; Wang et al., 2023), single-step transitions (Lu et al., 2023), subgoals (Black et al., 2024), trajectories (Janner et al., 2022; Ajay et al., 2023a; He et al., 2023; Carvalho et al., 2023; Luo et al., 2024), and videos for planning (Ajay et al., 2023b; Yang et al., 2024; Du et al., 2023; 2024). However, prior works primarily focus on single-domain generation tasks, such as data augmentation, without addressing cross-domain challenges. In contrast, xTED introduces a distinct setting by employing diffusion models to adjust source trajectories so that they more closely resemble target domain characteristics while preserving useful primitive information from source domains. As demonstrated in Section 5.5, this setting is highly sample-efficient, as it does not require the extensive training data needed by diffusion-based generation tasks that aim to generate data from modeled target distribution.

## 3 PRELIMINARIES

**Notations.** For convenience, we introduce the relevant notations and elements of decision trajectories with the standard formulation of Markov Decision Process (MDP). A finite-horizon MDP is defined as a tuple $\mathcal{M} := \langle \mathcal{S}, \mathcal{A}, T, R, H, \gamma \rangle$ where $\mathcal{S}$ and $\mathcal{A}$ are state and action spaces, $T : \mathcal{S} \times \mathcal{A} \to \Delta_{\mathcal{S}}$ is the transition dynamics; $R : \mathcal{S} \times \mathcal{A} \to \Delta_{\mathbb{R}}$ is the reward function; $H$ is the horizon that indicates the end of trajectories at the $H$-th step; $\gamma$ is the discount factor. The entire decision trajectory can be denoted with a sequence of transition tuples $\tau = \{(s_t, a_t, r_t)\}_{t=0}^{H-1}$, where $(s_t, a_t) \in \mathcal{S} \times \mathcal{A}$, $r_t \sim R(s_t, a_t)$ and $s_{t+1} \sim T(s_t, a_t)$. Nevertheless, trajectory modeling defined under these Markovian assumptions may result in limited performance as ablated in Section 5.6, as it enforces temporally causal relations within trajectories, potentially sacrificing global consistency.

**Source and Target Domains.** Adapting the standard MDP with domain-dependent annotations as $\mathcal{M}(\Omega) := \langle \mathcal{S}_\Omega, \mathcal{A}_\Omega, T_\Omega, R_\Omega, H, \gamma \rangle$ (Niu et al., 2024), where $\mathcal{S}_\Omega$, $\mathcal{A}_\Omega$, and $T_\Omega$ represent domain-dependent state spaces, action spaces, and transition dynamics, respectively. The reward function $R_\Omega$, influenced by $\mathcal{S}_\Omega$ and $\mathcal{A}_\Omega$, is also task-relevant. Here, $\Omega$ denotes the domain, which is shaped by the environment and the embodiments. Source domain(s) $\Omega_{src} = \{\Omega_{src}^i\}_{i=1}^N$, $N \in \mathbb{N}^+$, and target domain $\Omega_{tgt}$ may differ in state, action, and reward spaces, as well as in transition dynamics, reflecting the complex interplay of visual/viewpoint and dynamics gaps in the real world.

**Diffusion Models.** Diffusion models have emerged as a powerful class of generative models (Ho et al., 2020), which typically involves solving two stochastic differential equation (SDE) phases: the forward and the reverse processes. The forward process models the gradual addition of noise to the data, transforming the data distribution $p(\mathbf{x}_0)$ into a noised distribution over a series of discrete time steps $K$. This process can be described by the following equation:

$$q(\mathbf{x}_k|\mathbf{x}_{k-1}) = \mathcal{N}(\mathbf{x}_k; \sqrt{\alpha_{k-1}}\mathbf{x}_{k-1}, (1 - \alpha_{k-1})\mathbf{I}), \mathbf{x}_0 \sim q(\mathbf{x}_0), k \in [1, \cdots, K] \quad (1)$$

where $\alpha_k$ are the variance schedules. Given $\mathbf{x}_0$, we could sample the results of forward SDE at any noise step $k$:

$$\mathbf{x}_k \sim \mathcal{N}(\mathbf{x}_k; \mu_k, \sigma_k^2\mathbf{I}) := \mathcal{N}(\mathbf{x}_k; \sqrt{\alpha_1\alpha_2\cdots\alpha_k}\mathbf{x}_0, (1 - \alpha_1\alpha_2\cdots\alpha_k)\mathbf{I}) \quad (2)$$

The reverse process, on the other hand, involves learning to denoise the data, effectively reversing the forward process. It can be modeled as:

$$p_\theta(\mathbf{x}_{k-1}|\mathbf{x}_k) = \mathcal{N}(\mathbf{x}_{k-1}; \mu_\theta(\mathbf{x}_k, k), \Sigma_k) \quad (3)$$

where $\mu_\theta(\mathbf{x}_k, k)$ is the learnable mean of reverse conditional distribution. The training process involves optimizing the variational lower bound of the data likelihood, a process which encourages the model to accurately reconstruct the data from noise. This can boil down to simply minimizing the difference of the forward posterior and reverse conditional distribution $D_{KL}(q(\mathbf{x}_{k-1}|\mathbf{x}_k, \mathbf{x}_0)|p_\theta(\mathbf{x}_{k-1}|\mathbf{x}_k))$ across all denoising steps, where $q(\mathbf{x}_{k-1}|\mathbf{x}_k, \mathbf{x}_0) = $

$\mathcal{N}(\mathbf{x}_{k-1}; \mu_q(\mathbf{x}_k, \mathbf{x}_0), \Sigma_q)$. As the variance is only dependent on pre-defined $\alpha$ coefficients, we can construct the variance of the approximate reverse distribution $\Sigma_k$ also as $\Sigma_q = \sigma_q^2 \mathbf{I}$. Thus, $\mu_\theta(\mathbf{x}_k, k)$ is to clearly approach the forward process posterior mean $\mu_q(\mathbf{x}_k, \mathbf{x}_0)$:

$$\arg\min_\theta D_{KL}(q(\mathbf{x}_{k-1}|\mathbf{x}_k, \mathbf{x}_0) | p_\theta(\mathbf{x}_{k-1}|\mathbf{x}_k)) \Leftrightarrow \arg\min \frac{1}{2\sigma_q^2} \|\mu_\theta(\mathbf{x}_k, k), \mu_q(\mathbf{x}_k, \mathbf{x}_0)\|_2^2 \quad (4)$$

In practice, we adopt a further simplified surrogate loss instead of the distribution mean matching:

$$\mathcal{L}_\theta = \mathbb{E}_{k\in[1,\cdots,K],\mathbf{x}_0\sim q(\mathbf{x}_0),\epsilon\sim\mathcal{N}(\mathbf{0},\mathbf{I})}\|\epsilon - \epsilon_\theta(\mathbf{x_k}, k)\|^2 \quad (5)$$

where $\epsilon_\theta$ represents a parameterized network predicting the noise based on $\mathbf{x}_k$ and $k$. This remains identical to estimating the forward posterior mean, by adopting necessary re-parameterization (Ho et al., 2020). In our context, the data samples $\mathbf{x}$ are assigned with decision trajectories $\tau$.

# 4 xTED: A Cross-Domain Trajectory Editing Framework

In Section 4.1, we introduce the design philoso-phies behind model architecture and analyze why it is adept at handling the intricate hetero-geneity, dependencies and temporal consistency within trajectories than naïve implementations, as illustrated in Fig. 2. Next, we formalize how it effectively scaffolds the cross-domain trajec-tory editing paradigm, as detailed in Section 4.2.

## 4.1 Model Architecture

**Encoding and decoding design for decision making data.** States, actions, and rewards in MDP present fundamentally dissimilar notions, entailing distinct physical meanings and consis-tency properties. This naturally distinguishes the trajectory-based models from image-based models, which only considers spatial represen-tations among visual patches (Peebles & Xie, 2023) and simply treats every pixel homoge-neously in modeling designs (Ho et al., 2020). Thus, xTED encodes and decodes state, action, and reward sequences $\tau^s, \tau^a, \tau^r$ in trajectories $\tau$ separately. In every noise step $k$, each sequence is first compressed or enlarged into latent rep-

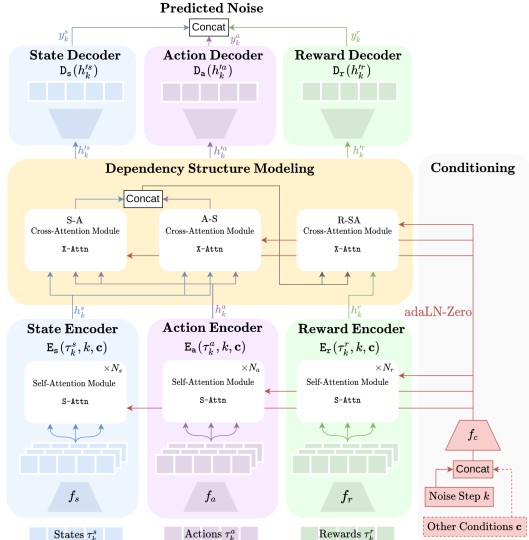

Figure 2: The model architecture is delicately de-signed for capturing intricate dependencies among transitions and decision elements (states, actions and rewards).

resentation $h_k^s, h_k^a, h_k^r$ with appropriate feature dimensions using separate networks $f$, and then processed with a self-attention module S-Attn to capture global consistency along the sequence dimension (conditions are also embedded with $\mathbf{e}_k^c = f_c(k, \mathbf{c})$):

$$h_k^i = \mathsf{E}_i(\tau_k^i, k, c) := \mathsf{S\text{-}Attn}(f_i(\tau_k^i), \mathbf{e}_k^c), i \in [s, a, r] \quad (6)$$

The series of networks $f$ provides substantial flexibility to incorporate task-specific prior knowledge, facilitating the construction of a reasonable information bottleneck. The embedding spaces should not be overly compressed, as subsequent attention blocks require sufficiently rich information to uncover dependencies. Conversely, these embedding spaces must not be excessively enlarged due to concerns about computational complexity and memory usage. From an interactive perspective, the embedding dimensions should be balanced to prevent the model from neglecting lower-dimensional action and reward embeddings when processing high-dimensional visual observations. At the end, given the final representation $h_k'^s, h_k'^a, h_k'^r$, we use separate decoders to predict the corresponding noise $y_k$ at noise step $k$, which recover the original dimension space respectively:

$$y_k^s = \mathsf{D}_s(h_k'^s) \in \mathbb{R}^\mathcal{S}, \ y_k^a = \mathsf{D}_a(h_k'^s) \in \mathbb{R}^\mathcal{A}, \ y_k^r = \mathsf{D}_r(h_k'^r) \in \mathbb{R}^R \quad (7)$$

Crucially, this separation on encoding and decoding preserves the inherent differences between states, actions, and rewards, aiding the model in recognizing clear dependencies on the latent representations $h_k$. Compared with prior works, these help avoid exploiting spurious correlations within decision-making data and sidesteps modeling the overall trajectory distribution directly (Janner et al., 2022; He et al., 2023), thereby enhancing sample efficiency and maximizing model expressiveness.

**Dependency structure modeling.** Although states, actions, and rewards possess fundamentally different physical properties, significant interactive dependencies exist among them. Specifically, unlike the spatial representation of images, sequential decision data entails intricate dynamics patterns that consistently drive state updates, action generation, and reward acquisition throughout trajectories, which play an indispensable role in effective trajectory modeling. To this end, the state and action embeddings $h_k^s$ and $h_k^a$ are cross-attended to capture mutual dependencies by exchanging key-value pairs in multi-headed attention, modeling the effects of policy and transition state dynamics separately:

$$h_k'^s = \text{X-Attn}(h_k^s, h_k^a, \mathbf{e}_k^c), \quad h_k'^a = \text{X-Attn}(h_k^a, h_k^s, \mathbf{e}_k^c) \tag{8}$$

However, it is important to note that rewards are naturally dependent on state-action sequences, while this dependency does not work in reverse due to their underlying causal relationships. Enforcing in-context optimality guidance could conflict with the primary goal of addressing domain gaps within trajectories. Therefore, we query the reward embeddings $h_k^r$ using concatenated state-action embeddings, without imposing unreasonable reward dependencies on state-action sequence modeling:

$$h_k'^r = \text{X-Attn}(h_k^r, [h_k^s, h_k^a], \mathbf{e}_k^c) \tag{9}$$

Introducing prior knowledge of the causal dependencies among states, actions, and rewards provides necessary regularizations for capturing trajectory distributions. This dependency structure modeling module alleviates the overabundance of information and relationships, enabling the model to accurately capture essential dependency priors and efficiently handle long-term sequential modeling. Additionally, the sequence-level dependency structure modeling naturally decouples the hypothesized temporal causal relationships adopted in many sequential modeling approaches (Chen et al., 2021; Ajay et al., 2023a), offering flexibility in whether to incorporate temporal causality during modeling.

**Incorporating external conditions.** In addition to the noise step $k$, our model architecture accommodates various conditioning information $\mathbf{c} = [c_i]_{i=0}^N$, guiding trajectories toward desirable regions. For example, we can steer the entire trajectory modeling toward high-rewarding regions by conditioning on normalized trajectory returns $\mathbf{c} = [R(\tau)]$ (Ajay et al., 2023a), with specific implementation details outlined in Appendix B.1. Moreover, our model demonstrates superior synergy with return-conditioned modeling compared to naive architectures that concatenate states, actions, and rewards along temporal or feature dimensions as input, as analyzed in Section 5.6. This advantage arises from our separate encoding-decoding manner and designs of dynamics dependency modeling, which effectively recognize the one-way dependencies between reward and state-action sequences.

**Applicability to diverse decision-making task settings.** Our architecture is compatible with multi-modal observations, including proprioceptive, exteroceptive (visual/point clouds), and language inputs, as long as modality-specific encoders are employed. From the perspective of trajectory composition, it offers flexibility in accommodating non-standard trajectories, which may include terminal flags, class labels, or preference labels for auxiliary module training to aid policy optimization. If these task-specific elements exhibit similarities with rewards, which have specific dependencies on states and/or actions, the architecture can be easily extended with minimal additional design complexities by incorporating extra encoders, decoders, and attention modules, akin to our reward-relevant design philosophies. Furthermore, practitioners can also omit the reward-relevant modules to fit non-reward (imitation learning) settings, as demonstrated in our real robot experiments (Section 5.4).

## 4.2 Cross-Domain Trajectory Editing

**Training Diffusion Model on Target Data.** Given a target domain dataset, we first train the diffusion model to capture its trajectory distribution using Eq. 5, where $\mathbf{x}_k$ represents the noised trajectory $\tau_k$, and additional conditions $\mathbf{c}$, if any, are concatenated with the noise step $k$ as inputs. Particularly, we keep the initial transition unchanged in each $H$-horizon trajectory during training, serving as an anchor for enhanced stability in long-sequence modeling.

**Editing Source Trajectories.** In the forward process, we obtain noised trajectories $\tau_k$ from $\mathcal{N}(\tau_k; \tau_0, \sigma_k^2 \mathbf{I})$, which retains information from the original source trajectory $\tau_0$. This procedure is governed by a ratio parameter $\kappa = \frac{k}{K} \in [0, 1]$, determining the level of noise addition. A small $\kappa$ preserves much of the source trajectory information, while a large $\kappa$ diminishes it. The ideal selection principle for $\kappa$ aims to distort only fine-grained information, such as transition dynamics in the source data, while retaining valuable large-scale information, such as skill primitives inherent in the trajectories. Additionally, this noised initialization theoretically yields favorable results by solving the reverse stochastic differential equation. Thus, we could then denoise $\tau_k$ using the diffusion prior,

successfully aligning the edited trajectories $\hat{\tau}_0$ with the target domain. The initial transition remains unchanged after each forward and denoising step, consistent with the training process, which is crucial for enhancing editing performance, as discussed in Appendix C.6. Experiments in Appendix C.2 indicate that setting $\kappa = 0.5$ is universally optimal for incorporating the edited source data into policy learning, ensuring sufficient exploitation of source data while shifting its domain properties toward the target domain. After the editing process, we can safely integrate edited source data with target data for downstream policy learning using any algorithms at task specifications.

## 5 EXPERIMENTS

In this section, we present empirical evaluations of the proposed cross-domain trajectory editing framework, xTED. We begin with detailing our experimental setups and the baselines for comparison. We then assess xTED against baseline methods in simulation (5.3) and real-robot tasks (5.4), overcoming various observation, dynamics, and morphology gaps within the source data. xTED consistently demonstrates advantages across different domain gaps and tasks compared to solely training on original target data or directly augmenting target data with source data (5.7), as well as its stability on various data quantity (5.5). We conclude with ablation studies on key design elements of our model architecture and provide evidence and analyses that echo with insights from methodology (5.6).

### 5.1 EXPERIMENTAL ENVIRONMENT SETUPS

**Simulation Experiments.** We conduct simulation-based experiments using the MuJoCo physics simulator (Todorov et al., 2012). Specifically, we construct two source domains on Walker2d-v2 (WK) and HalfCheetah-v2 (HC), where we intentionally introduce dynamics and morphology gaps relative to the original target domains by modifying the physics modeling configurations: (1) **Gravity**: applying $2\times$ gravitational acceleration in the simulation dynamics; (2) **Friction**: using $0.25/0.5\times$ friction coefficient to make the agent harder to maintain balance; (3) **Thigh Size**: using $2\times$ thigh size to introduce morphology gaps on the embodiment. We record the average return of Soft Actor-Critic (Haarnoja et al., 2018) policies trained in these source domains and provide a straightforward quantification of the impact of domain gaps on source policy learning in Appendix B.3.

For data acquirement and implementation details: (i) **target data**: we randomly select 20k transitions from the corresponding datasets in the standard offline RL benchmark D4RL (Fu et al., 2020). Specifically, we only consider the Medium (M), Medium Replay (MR) and Medium Expert (ME) datasets, as we hardly use a random or expert policy for system control in common real-world scenarios. (ii) **source data**: we collect trajectories of 20k transitions from the source domain with a SAC policy trained in the same domain. Please see Appendix B.1 for more implementation details.

**Real-Robot Experiments.** We conduct real-world experiments in robotic environments where target data is collected by the WidowX robot and source data is collected by the Airbot, for 100 trajectories respectively. We build three manipulation tasks: (1) **Pick up a red cup and place it on a silver pan (Cup)**; (2) **Pick up a duck and place it on a green plate (Duck)**; (3) **Move a pot from right to left (Pot)**. As shown in Fig. 3, there are huge domain discrepancies on robot embodiments and camera viewpoints: two 7-DoF robots possess distinctive module shapes/lengths, embodiment masses and joint types; source and target configurations present different viewpoints and field of views from base and wrist cameras on two robots. Our objective is to incorporate the edited data from the Airbot to enhance policy performance on the WidowX robot. These task environments are highly stochastic, featuring randomly initialized object positions and poses, as well as distractors with various shapes, colors, and locations. Please see Appendix B.2 for more implementation details.

### 5.2 BASELINES

(1) **Target**: training policies on the pre-sampled D4RL data (20k transitions per task) for simulation experiments and pre-collected WidowX data ($\sim$5k transitions per task) for real-robot experiments; (2) **Source**: training policies on data (20k transitions per task) pre-sampled with SAC policies for simulation experiments; (3) **Target with S4RL**: training policies solely on target data while applying the traditional data augmentation technique S4RL (Sinha et al., 2022) that adds action noise to target data; (4) **Target+Source**: training policies on both target data and unprocessed source data (20k transitions collected by SAC source policies for simulation experiments/around 6k transitions from human tele-operation on Airbot for real-robot experiments); (5) **Target+Edited Source** : training policies on both target data and source data edited with our pre-trained model model.

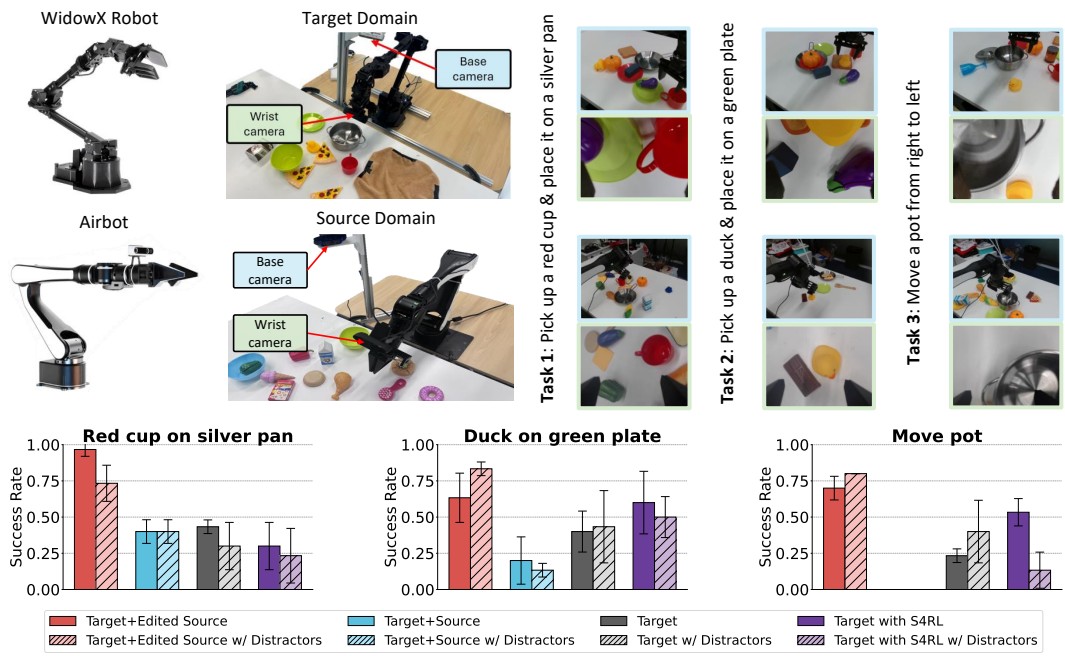

Figure 3: Target and source domains with complicated discrepancies on embodiments and viewpoints (top) and experiment results (bottom). The top right presents the snapshots from base and wrist camera views of data collection processes in target/source domain from **Cup/Duck/Pot** tasks respectively. The average success rate for real-robot tasks with/without distractors is obtained over 3 seeds.

## 5.3 COMPARATIVE EVALUATION IN SIMULATION EXPERIMENTS

In Table 1, we present the comparative results of xTED and other baselines on the simulation-based experiments. We integrate our trajectory editing process with a downstream policy learning module, implemented using one of the widely-used state-of-the-art (SOTA) offline RL algorithms, Implicit Q-Learning (IQL) (Kostrikov et al., 2022). Notably, our method achieves the best or on-par performance in almost all tasks (18 out of 18) compared to the baselines. Specifically, we find that policy learning directly augmented with original source data can have negative effects compared to solely learning from target data in 5 out of 18 tasks, while augmenting with edited source data consistently shows improvement in all of the tasks. In WK-MR and HC-MR, edited source data helps improve the performance by over 50% and 20% respectively. The overall 16.4% improvement also shows consistent stability on different domain gaps and quality of target data. This likely results from the fact that biased dynamics in source trajectories can detrimentally affect policy learning, despite the increased data coverage and diversified behavior patterns introduced by these trajectories. Nevertheless, xTED effectively adjusts the trajectories to realistic dynamics and provides meaningful augmentation, rather than merely expanding the data distribution. xTED also demonstrates the ability to handle vairous gaps simultaneously using a single fixed diffusion model in Appendix C.9.

## 5.4 COMPARATIVE EVALUATION IN REAL-ROBOT EXPERIMENTS

The average success rates for **Cup**, **Duck**, and **Pot** tasks are shown in Fig. 3. Each result is estimated over 10 episodes per task with random environmental initializations and averaged over 3 seeds. Target + Edited Source overwhelmingly outperforms the baselines in all the tasks, regardless of the presence of distractors. Notably, in the **Cup** task, combining learning from edited source data increases the success rate from 43% to 97% and 30% to 73.3%, in environments without and with distractors respectively. In contrast, due to the huge domain gaps between the source and target domains, directly adding original source data yields no performance gain and sometimes even causes degradation against solely training on target data, e.g. 40% to 20% in **Duck** task and 23% to 0% in **Pot** task. When evaluated with surrounding distractors, directly incorporating source data into policy learning sometimes leads to a more severe performance drop than without distractors, such as from 43% to 13% and from 40% to 0% in the **Duck** and **Pot** tasks, respectively. Furthermore, our method consistently outperforms the Target with S4RL across all tasks. This result highlights that

Table 1: Average normalized scores for MuJoCo tasks on 20k target/source data (5 random seeds). Δ denotes adding (edited) source data makes how much performance gain/degradation against training on target data.

| Target Data | | Source Dynamics | Src | Tgt | Tgt+Src | Δ | Tgt+Src (Edited) | Δ |
|---|---|---|---|---|---|---|---|---|
| Halfcheetah | Med | Gravity | 8.4±6.3 | 39.5±2.4 | **41.1±1.3** | +4.1% | **40.6±2.0** | +2.8% |
| | | Friction | 14.4±3.6 | 39.5±2.4 | 27.9±7.2 | -29.4% | **41.2±1.6** | +4.3% |
| | | Thigh Size | -0.1±1.0 | 39.5±2.4 | 40.3±2.2 | +2.0% | **40.7±2.4** | +3.0% |
| | Med-R | Gravity | 8.4±6.3 | 26.2±3.5 | 29.5±2.9 | +12.6% | **31.3±2.9** | +19.5% |
| | | Friction | 14.4±3.6 | 26.2±3.5 | 23.2±4.0 | -11.5% | **31.8±3.1** | +21.4% |
| | | Thigh Size | -0.1±1.0 | 26.2±3.5 | 29.0±3.7 | +10.7% | **33.0±3.0** | +26.0% |
| | Med-E | Gravity | 8.4±6.3 | 40.1±2.9 | 40.4±3.4 | +0.7% | **43.8±3.6** | +9.2% |
| | | Friction | 14.4±3.6 | 40.1±2.9 | 27.1±5.4 | -32.4% | **43.2±3.0** | +7.7% |
| | | Thigh Size | -0.1±1.0 | 40.1±2.9 | 37.4±5.5 | -6.7% | **43.0±3.0** | +7.2% |
| Walker2d | Med | Gravity | 16.6±6.2 | 45.3±15.9 | **60.4±9.5** | +33.3% | 58.2±11.7 | +28.5% |
| | | Friction | 8.2±1.4 | 45.3±15.9 | 47.0±12.8 | +3.8% | 54.5±13.7 | +20.3% |
| | | Thigh Size | 7.6±3.5 | 45.3±15.9 | 49.6±12.2 | +9.5% | **58.9±11.7** | +30.0% |
| | Med-R | Gravity | 16.6±6.2 | 16.6±5.9 | 19.5±10.7 | +17.5% | **23.3±9.1** | +40.3% |
| | | Friction | 8.2±1.4 | 16.6±5.9 | 17.4±6.2 | +4.8% | **25.9±9.1** | +56.0% |
| | | Thigh Size | 7.6±3.5 | 16.6±5.9 | 18.0±6.7 | +8.4% | **25.9±9.1** | +56.0% |
| | Med-E | Gravity | 16.6±6.2 | 71.0±21.0 | 67.4±11.1 | -16.9% | **82.9±18.1** | +16.8% |
| | | Friction | 8.2±1.4 | 71.0±21.0 | **75.1±18.3** | +5.8% | 74.0±24.5 | +4.2% |
| | | Thigh Size | 7.6±3.5 | 71.0±21.0 | 77.0±19.3 | +8.5% | **81.0±21.4** | +14.1% |
| | | Total | 165.3 | 716.1 | 727.3 | +1.6% | **833.2** | +16.4% |

Table 2: Reduced data usage for diffusion training hardly has negative impact on policy learning.

| Domain Gap | Gravity | | | Friction | | | Thigh Size | | |
|---|---|---|---|---|---|---|---|---|---|
| Target Data | 5k | 10k | 20k | 5k | 10k | 20k | 5k | 10k | 20k |
| Avg. Score | **26.7±9.8** | 21.5±8.5 | 23.3±9.1 | **29.0±8.3** | 25.0±8.2 | 25.9±9.1 | **31.4±10.0** | 29.8±9.6 | 25.9±9.1 |

data augmentation methods cannot reliably enhance policy performance when compared to using only the original target data. It underscores the superior effectiveness and necessity of the cross-domain trajectory editing paradigm, which appropriately leverages biased source data. All numeric results in Fig. 3 can be found in Appendix C.1 Table 7.

## 5.5 INVESTIGATIONS ON SAMPLE EFFICIENCY

xTED demonstrates robust performance across varying amounts of data. We first ablate on data quantity for diffusion training (5k, 10k, 20k transitions from WK-MR) and keep the same data quantity for policy learning. From Table 2, we find that the policies learned from target and edited source data still perform well despite of very limited data for training diffusion model, demonstrating high sample efficiency of diffusion models in our data editing paradigm.

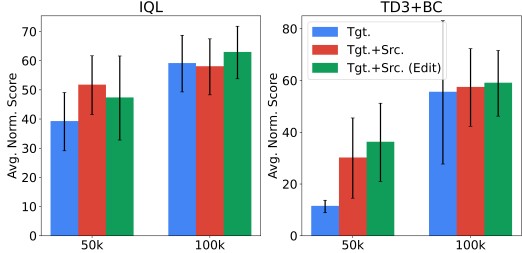

Figure 4: Different amounts of data for diffusion and policy training (Target: WK-MR, Source: Gravity).

We then ablate on data quantity for both diffusion training and policy learning: randomly sample transitions of 50k and 100k from the WK-MR dataset for diffusion training, combining each with an equivalent amount of source data for policy learning. Fig. 4 exhibits the policy performance on target, target+source, and target+edited source datasets, utilizing different training algorithms, IQL and TD3+BC (Fujimoto & Gu, 2021). It is evident that with small datasets (20k transitions, Table 1), our method significantly enhances the quality of the source domain dataset, thereby improving training performance. As the dataset size increases (50k and 100k transitions), our method continues to enhance source data quality, though the improvement over baselines might sometimes become less pronounced since the target data itself can sufficiently support policy learning without source data. xTED also prevails on larger amounts of source data involved in policy learning in Appendix C.4.

Table 3: Architecture Ablations (5 random seeds). Red indicates lower scores than solely training on target data.

| Target Data | Source Dynamics | TC | FC | Inv | CM | Ours |
|---|---|---|---|---|---|---|
| HC-MR | Gravity | 23.8±3.2 | 25.6±3.1 | 28.8±3.4 | 20.1±3.2 | **31.3±2.9** |
| | Friction | 21.9±3.7 | 22.4±4.0 | 27.9±3.6 | 13.1±3.4 | **31.8±3.1** |
| | Thigh Size | 27.6±3.5 | 26.6±4.1 | 30.9±2.7 | 19.3±2.1 | **33.0±3.0** |
| WK-MR | Gravity | 23.1±9.8 | 15.9±6.9 | 20.8±6.3 | 11.4±5.1 | **23.3±9.1** |
| | Friction | 20.4±6.6 | 20.7±8.5 | 20.7±7.6 | 11.4±5.4 | **25.9±9.1** |
| | Thigh Size | 20.8±7.5 | 20.3±6.2 | 20.3±9.0 | 9.6±4.0 | **25.9±9.1** |
| | Total | 137.6 | 131.5 | 149.4 | 84.9 | **171.2** |

Table 4: Our architecture is more compatible with (return-) conditioned diffusion. Results are averaged over 5 random seeds. Red indicates lower scores than solely training on target data. ↑ indicates improved performance with return conditioning, while ↓ indicates the opposite.

| Target Data | Source Dynamics | TC | (Return Cond.) | FC | (Return Cond.) | Ours | (Return Cond.) |
|---|---|---|---|---|---|---|---|
| HC-MR | Gravity | 23.8±3.2 | 21.3±6.2 ↓ | 25.6±3.1 | 19.6±6.5 ↓ | **31.3±2.9** | 30.5±3.1 ↓ |
| | Friction | 21.9±3.7 | 23.8±5.6 ↑ | 22.4±4.0 | 20.1±7.9 ↓ | 31.8±3.1 | **35.1±1.8** ↑ |
| | Thigh Size | 27.6±3.5 | 21.1±7.7 ↓ | 26.6±4.1 | 24.9±6.4 ↓ | 33.0±3.0 | **33.3±3.0** ↑ |
| WK-MR | Gravity | 23.1±9.8 | 8.9±5.2 ↓ | 15.9±6.9 | 7.5±4.2 ↓ | 23.3±9.1 | **24.5±9.0** ↑ |
| | Friction | 20.4±6.6 | 9.0±5.5 ↓ | 20.7±8.5 | 7.2±3.8 ↓ | **25.9±9.1** | 22.3±8.7 ↓ |
| | Thigh Size | 20.8±7.5 | 11.9±6.3 ↓ | 20.3±6.2 | 10.0±4.8 ↓ | 25.9±9.1 | **31.1±11.5** ↑ |
| | Total | 137.6 | 96.0 ↓ | 131.5 | 89.3 ↓ | **171.2** | **176.8** ↑ |

## 5.6 Ablation Findings on Model Architecture

We ablate our model architecture to demonstrate the impact of different design insights and techniques. We primarily consider four distinct architecture baselines: **(1) Feature Concatenation (FC):** concatenating state, action, and reward sequences on feature dimension before passing through attention; **(2) Transition concatenation (TC):** concatenating state, action, and reward sequences on temporal dimension before passing through attention; **(3) Causal Mask (CM):** incorporating causal masks in attention modules in our architecture for modeling Markovian process, considering transitions only depending on the previous step; **(4) Diffusing over States + Inverse Dynamics (Inv):** solely diffusing over state trajectories, while employing two inverse models to obtain actions and rewards. Table 3 demonstrates that our model consistently achieves superior performance, from which several key insights into architecture design can be derived: (1) *Entire trajectory modeling outperforms partial trajectory modeling*: Using inverse models (Inv) results in significantly higher compounding errors, as action feasibility and reward plausibility cannot be reliably maintained when only states are diffused without a unified model that handles all components consistently; (2) *Imposing Markovian properties as priors in trajectory modeling is suboptimal*: Incorporating Markovian modeling (CM) degrades the quality of edited source data, as local transition dynamics enforced by masked attention mechanisms may conflict with the global temporal consistency required in trajectory modeling. This suggests that trajectory modeling demands a distinct approach compared to common practices in decision-making approaches; (3) *Our design effectively captures intricate dependencies within trajectories*: Concatenating states, actions, and rewards along transition or feature dimensions (TC/FC) is problematic. It treats the entire trajectory as a homogeneous, image-like input, which obscures the model's ability to correctly identify dependencies between transitions and decision elements, and introduces unwarranted dependencies of rewards on state-action sequences; (4) *Our design integrates well with return-conditioned diffusion*: Table 4 reveals that incorporating return conditioning in naïve architectures (TC/FC) leads to performance degradation, whereas our model with return conditioning successfully guides edited source data towards higher-reward regions, resulting in superior policy performance. This is because only our design correctly models the one-way dependencies of reward and state-action sequences, enabling effectively implementing return-conditioned strategies.

## 5.7 xTED for Single-Domain Data Augmentation

In reality, data from other domains is not always available. Fortunately, our model design also entails great potential to serve as a data generation model to navigate through this situation. In Fig. 5, we illustrate the results of using our model for data augmentation on small datasets (20k transitions) pre-sampled from D4RL. Our model offers two simple types of augmentation approaches: **(1) Tgt. (Edit)**: adding noise onto the original data as the forward process to match a theoretical Gaussian

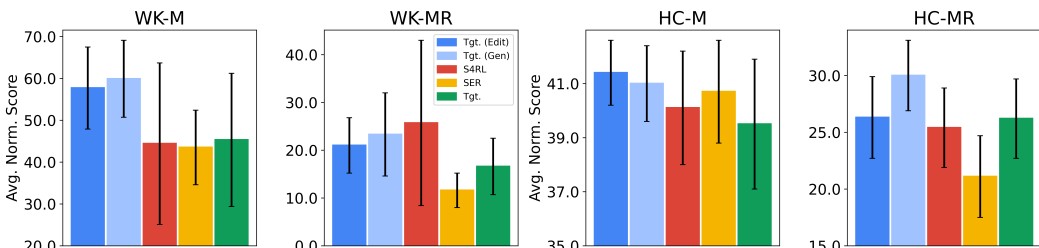

Figure 5: Average normalized scores for single-domain data augmentation. Averaged over 5 random seeds.

distribution ($\kappa = 1.0$), followed by denoising with our model with the same number of steps. **(2) Tgt. (Gen)**: directly denoising standard Gaussian noise with our model. model-based trajectory augmentation is compared with two baselines: **(1) S4RL** (Sinha et al., 2022): adding a zero-mean noise distribution $\mathcal{N}(0, 3e^{-4})$ on every dimension of the state space; **(2) SER** (Lu et al., 2023): learn a diffusion-based dynamics model for transition generation. All the methods augment the original dataset with the same amount of synthetic samples. It can be observed that augmentation with our diffusion model consistently leads in almost all tasks. S4RL achieves comparable scores in specific tasks but tends to exhibit high variance due to its randomization mechanism. Compared with SER, our approach demonstrates that modeling longer-horizon trajectory dynamics and temporal consistency significantly improves data augmentation quality for small datasets over merely modeling transition dynamics.

## 6 CONCLUSIONS

We present a novel, simple, generic, flexible, and effective trajectory editing paradigm (xTED), which reframes cross-domain policy adaptation as a data pre-processing problem. It is a task- and domain-agnostic method that avoids the need for task-specific policy adaptation designs and can accommodate multiple available source domains. Furthermore, it is compatible with any observation encoder and downstream policy learning method, making it adaptable to various task specifications. Additionally, xTED is orthogonal to other cross-domain policy adaptation paradigms, which can be flexibly integrated for greater synergy if necessary. At the core of xTED, we introduce a scalable and flexible diffusion model architecture, effectively modeling the complex dependencies within trajectories. This design proves to be sample-efficient and highly compatible with (return-)conditioned diffusion. Besides, our model demonstrates strong potential as a data generator for single-domain data augmentation when no source data is available, highlighting its superiority in modeling entire trajectories. Through extensive experiments, we show that incorporating source data edited with xTED consistently yields performance improvements over training solely on target data while directly adding unprocessed source data often results in significant performance degradation, particularly in real-robot manipulation tasks.

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

APPENDIX

## A    EXTENDED DISCUSSIONS WITH EXISTING WORKS

xTED offers a generic, flexible, simple yet effective cross-domain trajectory editing paradigm that gains massive necessity and advantages over existing approaches. This section comprehensively elaborates on its distinctions from domain adaptation and data augmentation.

### A.1    DOMAIN ADAPTATION

xTED casts cross-domain policy adaptation as a data pre-processing problem, thereby eliminating the need for designing modality/task-specific adaptation modules that function during training. Most existing policy adaptation methods primarily focus on achieving **MDP Alignment** (Kim et al., 2020).

**Existing categories of domain adaptation.** The first category of approaches to achieving such alignment is through **learning direct mappings or corrections** between source and target domains. For observation space alignment, numerous works leverage advances in generative adversarial learning and contrastive learning to transform the observation space, achieving consistent visual rendering and camera viewpoints as in the target domain (Zhu et al., 2017; Liu et al., 2018; Zhang et al., 2019; Bewley et al., 2019; Rao et al., 2020; Liu et al., 2020; Gangwani & Peng, 2020; Radosavovic et al., 2021). In scenarios with dynamics mismatch, it becomes necessary to align both state and action spaces simultaneously, which introduces more complex correspondence learning mechanisms such as (dynamics/temporal) cycle consistency (Kim et al., 2020; Zhang et al., 2020; Raychaudhuri et al., 2021; Zakka et al., 2021; Wang et al., 2022b) and mutual information criteria (Franzmeyer et al., 2022). Alternatively, modifying the reward space can compensate for the misalignment in transitional dynamics within MDPs (Eysenbach et al., 2021; Liu et al., 2022; Lyu et al., 2024; Xue et al., 2023), which may also be implicitly addressed by adaptively regularizing Q-values (Niu et al., 2022; 2023; Hou et al., 2024). *This category necessitates model retraining when new data from other source domains becomes available, which can be time-consuming and labor-intensive.*

The second category resorts to **building a canonical representation space** that preserves task-relevant yet domain-agnostic information (Stadie et al., 2016; Pan et al., 2017; Mueller et al., 2018; Sermanet et al., 2018; Cetin & Celiktutan, 2021; Wang et al., 2022a; Yang et al., 2023; Choi et al., 2024). This auxiliary representation space, often viewed as skills (Gupta et al., 2017; Hejna et al., 2020; Pertsch et al., 2022; Xu et al., 2023b) or subgoals (Sharma et al., 2019; Ma et al., 2022; 2023; Li et al., 2024), facilitates the construction of aligned domain-invariant MDPs by capturing task semantics that effectively guide agents in completing tasks. *However, this approach involves designing task-specific loss functions and optimization procedures, which may struggle to find a representation space that effectively encapsulates the inputs and reflects the task.*

Despite the aforementioned shortcomings, these two categories share a common Achilles' heel: mode collapse (Lopez et al., 2021). Most policy adaptation methods, whether explicitly or implicitly, rely on the high-level philosophy of generation and discrimination. *Learning an informative mapping or a canonical representation that can effectively recover the entire target distribution can be exceedingly challenging, especially due to data imbalance and the sensitivity of hyper-parameter tuning.*

**Why "data adaptation"?** To overcome all the aforementioned issues, it is crucial to avoid designing modality/task-specific adaptation modules and instead **handle domain adaptation at the data level**. This approach provides significant flexibility, allowing the integration of various upstream observation encoders for different modalities, as well as diverse downstream policy learning methods that best suit the tasks at hand.

**Why diffusion-based data editing?** Direct data transformation cannot rely on a pre-trained deterministic mapping or representation to pre-process source data, as this would face the same challenges as traditional domain adaptation methods, leading to transformed data with reduced variety and insufficient coverage in the target distribution. Moreover, such mappings or representations require re-training or fine-tuning to accommodate multiple source domains. In contrast, diffusion-based editing ensures diversity in the transformed data due to its stochastic nature and avoids mode collapse. Additionally, the editing process is universally domain-agnostic, correcting only the dynamics and observation patterns while preserving task-relevant information in source data with a target distribution prior modeled by a diffusion model, thus enabling a retraining-free approach.

Above all, xTED represents a **novel, generic, and flexible** cross-domain data-level policy adaptation paradigm. It is *domain-agnostic*, accommodating multiple source domains without the need for re-training or fine-tuning the diffusion model; *modality-agnostic*, being compatible with various observation encoders; *task-agnostic*, integrating seamlessly with different task-specific policy learning methods; *compatible* with other policy adaptation paradigms and approaches for greater synergy.

## A.2 DISTINCTIONS FROM DATA AUGMENTATION

**Setting.** Data augmentation is primarily employed in single-domain settings, which are fundamentally different from xTED's cross-domain applications. Typically, no data from other domains is introduced, and data augmentation focuses on manipulating training data to enhance robustness, e.g. noising the original data (Sinha et al., 2022).

In contrast, data editing is a broader paradigm that encompasses the applications of data augmentation. Even when data from other domains is unavailable, xTED can serve as an expressive data generation model for augmenting small-sample datasets in single-domain settings, outperforming state-of-the-art data augmentation strategies (Lu et al., 2024; Sinha et al., 2022) (see Section 5.7).

**Diffusion-based data editing V.S. data augmentation.** We have to emphasize that using diffusion models for data editing is both sample-efficient and error-tolerant than diffusion-based trajectory generation for data augmentation (Lu et al., 2023; He et al., 2023; Jackson et al., 2024). This is because the goal in data editing is far more easily attainable compared to its application in data generation tasks. When diffusion models are used for data generation, the focus is usually on achieving a high level of precision to generate data that closely matches the original data distribution. However, in the context of data editing, the requirements are relaxed to simply distort (edit) the data to more closely match the target dynamics and observation patterns than the source data. Crucially, there is no need for edited source data to closely resemble target data in every detail. Thus, diffusion models for editing are relatively insensitive to the amount of target data used, having minimal impact on the ultimate policy performance as proved in Section 5.5.

# B IMPLEMENTATION DETAILS AND EXPERIMENT SETUP

## B.1 SIMULATION EXPERIMENTS

The implementation details in our simulation experiments are specified as follows:

- **Model and Method Setups**: to strike the balance among the concerns from Separate Encoding-Decoding in Section 4.1, we try to encode the state, action, and reward into $16\times$ embedding dimensions respectively. In terms of the number of self-attention blocks, we set $N_s = N_a = 2, N_r = 1$ in Figure 2 for the least computation consumption with performance guarantee. The training process takes denoising steps $K = 200$ and optimizes over Eq. 5. In xTED, we perturb the source trajectories with $k = 100$ steps of gaussian noises and denoise them with the pretrained model in $k = 100$ steps ($\kappa = 0.5$).

  The computation of attention modules is proportional to $9H^2$ (self-attention: $(N_s + N_a + N_r)^2$; cross-attention: $4H^2$), so we also set architecture with minimal differences on computation efforts for the architecture ablation baselines for fair comparisons, i.e. FC, TC, Inv, CM. For CM, we keep the architecture the same with xTED, with the only difference of adding a causal mask with markovian constraints. We use 9 successive self-attention modules and FC ($9 \times H^2$), while one self-attention module for TC ($(3 \times H)^2$), achieving the same attention computation. For Inv, we only have to diffuse over states without addressing the intricate dependencies within trajectories so we reduce the computation and adopt only 2 self-attention modules for efficiency.

- **Data Usage**. We use only 20k transitions from the D4RL target dataset for diffusion training and policy learning on each task, which represents merely 1/50 of HC-M, 1/10 of HC-MR, 1/100 of HC-ME, 1/50 of WK-M, 1/15 of WK-MR, and 1/100 of WK-ME from the original D4RL dataset. The full dataset is already sufficient to train a high-performing policy, making it unnecessary to combine source data and risk potential negative effects due to domain gaps.

- **Encoder-Decoder**. Separately encoding and decoding state, action, and reward sequences in trajectories offers great flexibility to apply task-specific prior knowledge in constructing a reasonable information bottleneck. The spaces of embedded state $z_s$, action $z_a$, and reward $z_r$ should not be overly compressed as following attention blocks requires sufficiently rich information to

discover connections. On the other hand, the embedding spaces should not be infinitely enlarged due to concerns on computational complexity and memory use. From an interactive angle, the sizes of these embedding spaces should not be diversely imbalanced so that the model is less prone to ignoring the low-dimensional action and reward embeddings when involved with visual observations. In all the transformer architectures ablated in Section 5.6, the encoders and decoders we used are simple MLPs with two dense layers:

$$
\begin{aligned}
z_s &= \text{MLP}(s) \\
z_a &= \text{MLP}(a) \\
z_r &= \text{MLP}(r) \\
\mathbf{z} &= [z_s, z_a, z_r]
\end{aligned}
\tag{10}
$$

In xTED and CM, we expand the observation, action, and reward into a feature (hidden) space with 16 times the original dimensionality. In Inv, FC, and TC, each observation, action, and reward is embedded into a 320-dimensional space.

- **Self-Attention Module S-Attn**$(x, c)$. In xTED, TC, FC, Inv and CM, we use multi-head self-attention (MSA) mechanism to apply long-horizon consistency in sequence modeling. Specificlly, we use the `MultiHeadDotProductAttention` (MSA) function implemented by JAX, denoted by ATTN. The forward processes of self-attention module are formulated as follows:

$$
\begin{aligned}
z_c &= \text{MLP}(c) \\
q = k = v &= \text{LayerNorm}(x) \cdot (1 + z_{c,\text{scale}}) + z_{c,\text{shift}} \\
x &= x + z_{c,\text{gate,attn}} \cdot \text{ATTN}(q, k, v) \\
x &= x + z_{c,\text{gate,mlp}} \cdot \text{MLP}(\text{LayerNorm}(x) \cdot (1 + z_{c,\text{scale}}) + z_{c,\text{shift}})
\end{aligned}
\tag{11}
$$

Here, $x$ is the input, $c$ is the condition, $z_{c,\text{gate}}$, $z_{c,\text{scale}}$ and $z_{c,\text{shift}}$ are chunked from $z_c$. More detailed hyperparameters are shown in Table 5.

- **Cross-Attention Module X-Attn**$(x_q, x_{kv}, c)$. In xTED and CM, we use cross-attention mechanism to apply sequence modeling. The input argument $c$ is produced by the output of the condition encoder $f_c$, which splits into two components, $c_q$ and $c_{kv}$, ensuring that the tensor shapes align with $x_q$ and $x_{kv}$, respectively. We still use the `MultiHeadDotProductAttention` implemented by JAX. The forward processes of cross-attention module are formulated as follows:

$$
\begin{aligned}
c_q, c_{kv} &= c \\
z_{c_q} &= \text{MLP}(c_q) \\
z_{c_{kv}} &= \text{MLP}(c_{kv}) \\
q &= \text{LayerNorm}(x_q) \cdot (1 + z_{c_q,\text{scale}}) + z_{c_q,\text{shift}} \\
k = v &= \text{LayerNorm}(x_{kv}) \cdot (1 + z_{c_{kv},\text{scale}}) + z_{c_{kv},\text{shift}} \\
x_{kv} &= x_{kv} + z_{c_{kv},\text{gate,attn}} \cdot \text{ATTN}(q, k, v) \\
x_{kv} &= x_{kv} + z_{c_{kv},\text{gate,mlp}} \cdot \text{MLP}(\text{LayerNorm}(x_{kv}) \cdot (1 + z_{c_{kv},\text{scale}}) + z_{c_{kv},\text{shift}})
\end{aligned}
\tag{12}
$$

Here, $x_q$ is the input for query, $x_{kv}$ is the input for key and value, $c_q$ is the condition for $x_q$, $c_{kv}$ is the condition for $x_{kv}$. $z_{c_{kv},\text{gate}}$, $z_{c_{kv},\text{scale}}$ and $z_{c_{kv},\text{shift}}$ are chunked from $z_{c_{kv}}$. $z_{c_q,\text{scale}}$ and $z_{c_q,\text{shift}}$ are chunked from $z_{c_q}$. More detailed hyperparameters are shown in Table 5.

- **Editing Details**. When we use xTED for trajectory editing, the step parameter $\kappa$ we use is 0.5, while the total time steps $K$ we used for training xTED model is 200. This implies that we add $k = 100$ steps of noise to the original source-domain data, then denoise it with xTED with the same number of steps.

- **Policy Learning**. In this paper, we use IQL and TD3+BC as policy learning algorithms. The key hyperparameters we used are shown in Table 5, which are basically the same as the original papers.

- **Evaluation**: All the simulation results are obtained by averaging over the same set of 5 random seeds.

- **Computing Resources**: we ran experiments largely on 8 NVIDIA A100 GPUs via an internal cluster.

Table 5: Hyperparameters for simulation (sim) and real-robot (real) experiments.

| Hyper-parameter | Value |
|---|---|
| **xTED** (sim) | |
| Number of self attention blocks for state | 2 |
| Number of self attention blocks for action | 2 |
| Number of self attention blocks for reward | 1 |
| Number of cross attention blocks for s-a | 1 |
| Number of cross attention blocks for a-s | 1 |
| Number of cross attention blocks for r-sa | 1 |
| Size of hidden dim for encoder/decoder | 16×original dim |
| Trajectory horizon $H$ | 20 |
| **xTED** (real) | |
| Number of self attention blocks for state | 2 |
| Number of self attention blocks for action | 2 |
| Number of cross attention blocks for s-a | 1 |
| Number of cross attention blocks for a-s | 1 |
| Size of hidden dim for encoder/decoder | 1×original dim |
| Trajectory horizon $H$ | 5 |
| **IQL** (sim) | |
| Actor learning rate | 3e-4 |
| Critic learning rate | 3e-4 |
| Optimizer | adam |
| **TD3+BC** (sim) | |
| Actor learning rate | 1e-5 |
| Critic learning rate | 1e-3 |
| Optimizer | adam |
| **BC** (real) | |
| Actor learning rate | 1e-4 |
| Optimizer | adam |

## B.2 REAL-ROBOT EXPERIMENTS

In addition to configurations identical to those used in simulation experiments, we summarize below the specific implementation details for the real-robot experiments:

- **Model and Method Setups**: Different from simulation experiments, the robot manipulation tasks involves environments where rewards are hard to specify, thus resulting in collecting non-reward trajectories. Correspondingly, the self-attention and cross-attention modules for reward trajectories are removed from the original xTED architecture, leaving only self-attention and cross-attention modules for state and action sequences. In terms of other architecture designs and editing details, we keep them identical with simulation experiment setups. We set the trajectory horizon for modeling with xTED to $H = 5$, as the robot manipulation trajectories contain higher-dimensional features, necessitating a trade-off in horizon length to reduce computational burden.

- **Data Collection**. Each source/target dataset comprises 100 trajectories (40-60 transitions per trajectory with a frequency of around 5Hz, 75 trajectories w/ distractors and 25 trajectories w/o distractors) with RGB observations (480 × 640) from an external camera and a camera mounted on the robot's gripper, as well as end-effector actions.

- **Observation Pre-processing**. We choose DecisionNCE-T (Li et al., 2024) to compress the 480x640 camera views into embedding space with 1024 feature dimensions before applying xTED. xTED is then deployed on the lower-dimensional observation representation instead of the high-dimensional image observations.

- **Encoder-Decoder.** In xTED architecture, we maintain the original feature dimensionality of the observation representation (1024) and expand the action space by two times orginal feature dimensionality.

- **Transformer Architecture.** The architecture remains identical to that used in our simulation experiments, with the only exception being the removal of modules related to reward dependency modeling.

- **Policy Learning**. The robot policy network consists of three layers of MLPs for action prediction, trained via vanilla behavior cloning (BC) with L2 regression using the Adam optimizer at a learning rate of 1e-4. All models are trained for 2.5k epochs with a batch size of 64.

- **Evaluation**. We evaluate the policy checkpoint with 10 episodes for each seed with random environmental initializations and 3 random seeds for each result.

- **Computing Resources**: we ran experiments on 4x NVIDIA A800 GPUs via an internal cluster.

### B.3 ON THE MAGNITUDE OF DOMAIN GAPS INTRODUCED BY THE SIMULATION EXPERIMENT SETUPS

In Section 5.1, we mentioned that we modified the original dynamics—such as 2x gravity, 0.25/0.5x friction (HC/WK), and 2x thigh size—to construct source domains with biased dynamics. To quantify the domain gaps, we recorded the mean values of the average returns of SAC policies trained in the source domain but evaluated in the target domain, and compared them to SAC policies trained and evaluated in the target domain. As shown in Table 6, the domain gaps significantly hinder direct policy transfer from the source to the target domain, resulting in notable performance degradation.

Table 6: Average scores of SAC policies trained in source/target domain and evaluated in target domain.

| Task | Domain Gap | Avg. Return (Source) | Avg. Return (Target) |
|------|-----------|----------------------|----------------------|
| Halfcheetah | Gravity | 4513 | |
| | Friction | 5099 | 10290 |
| | Thigh Size | 5537 | |
| Walker2d | Gravity | 1233 | |
| | Friction | 3392 | 4916 |
| | Thigh Size | 437 | |

## C ADDITIONAL EXPERIMENT RESULTS

### C.1 NUMERIC RESULTS OF REAL-ROBOT EXPERIMENTS

The values of success rate depicted in Fig. 3 bar plots are listed in the following Table 7. Surprisingly, incorporating edited source trajectories helps improve success rate of task completion by a fairly large margin, i.e. $> 100\%$ or even $> 200\%$ against solely training on target data. However, directly combining original source trajectories generally results in severe performance degradation, i.e. in 5/6 tasks. Particularly, the success rates drop to 0% in Pot tasks where the robot arm cannot identify the target object and approach it.

Qualitatively, as shown in the recorded videos on our website, robot policies trained on target data consistently exhibit poor performance in lifting the end effector, preventing the cup from being placed through the top opening of the pan. Additionally, these policies show low accuracy and poor timing in grasping or releasing target objects, attributed to the insufficiency and narrowly distributed target data. Conversely, directly incorporating source data results in policies that are more aggressive and less precise in control, as Airbot operates within a broader range of control dynamics.

### C.2 XTED WITH DIFFERENT CHOICES OF EDITING RATIO

According to SDEdit (Meng et al., 2022), editing ratio $\kappa = 0.5$ falls in the sweet spot of image editing tasks and so does it in trajectory editing tasks in general, as shown in Table 8. $\kappa = 0.0$ indicates no editing applied on the source data, whereas $\kappa = 1.0$ indicates that edited source data is theoretically denoised from random noise and thus approximately equivalent to direct generation without any basis. This reveals that $\kappa$ should not be too large or too small to effectively retain useful information from the source data while precisely eliminating domain-specific biases. However, in certain scenarios, $\kappa = 1.0$ can successfully transform source trajectories to align with target domain properties, resulting in the best performance.

Table 7: Performance comparisons across different real-robot tasks and baselines.

| Task | Target+Edited Source (Ours) | Target+Source | Target | Target (w/ S4RL) |
|------|------------------------------|----------------|--------|------------------|
| Cup | 0.97±0.06 (+125.6%) | 0.40±0.10 (-7.0%) | 0.43±0.06 | 0.30±0.20 |
| (w/ distractors) | 0.73±0.15 (+143.3%) | 0.40±0.10 (+33.0%) | 0.30±0.20 | 0.23±0.23 |
| Duck | 0.63±0.21 (+57.5%) | 0.20±0.20 (-50.0%) | 0.40±0.17 | 0.60±0.26 |
| (w/ distractors) | 0.83±0.06 (+93.0%) | 0.13±0.06 (-69.8%) | 0.43±0.31 | 0.50±0.17 |
| Pot | 0.70±0.10 (+204.3%) | 0.00±0.00 (-100.0%) | 0.23±0.06 | 0.53±0.11 |
| (w/ distractors) | 0.80±0.00 (+100.0%) | 0.00±0.00 (-100.0%) | 0.40±0.26 | 0.13±0.15 |

Table 8: Ablations on editing ratio $\kappa$ with WK-MR dataset.

| Domain Gap | $\kappa$ | Target+Edited Source |
|------------|----------|----------------------|
| Gravity | 0.0 | 19.5±10.7 |
| | 0.05 | 23.1±10.4 |
| | 0.5 | 23.3±9.1 |
| | 1.0 | 27.1±10.8 |
| Friction | 0.0 | 17.4±6.2 |
| | 0.05 | 24.7±7.0 |
| | 0.5 | 25.9±9.1 |
| | 1.0 | 25.5±13.6 |
| Thigh Size | 0.0 | 18.0±6.7 |
| | 0.05 | 25.7±7.9 |
| | 0.5 | 25.9±9.1 |
| | 1.0 | 24.1±9.3 |

## C.3 XTED WITH MULTIPLE EDITING ITERATIONS

In xTED, we add $k$-step Gaussian noise to source trajectories and denoise them in $k$-steps using a pre-trained xTED for one iteration ($e = 1, \kappa = 0.5$ for all our results). In Table 9, we explore the impact of two editing iterations ($e = 2$) and find that increasing the number of iterations can sometimes enhance xTED's performance by achieving better alignment with target domain properties. It is noteworthy that $e = 1, \kappa = 1.0$ and $e = 2, \kappa = 0.5$ involve the same total number of noising and denoising steps, with the only difference being that $e = 2, \kappa = 0.5$ divides the process into two iterations. $e = 2, \kappa = 0.5$ seems to perform well only if $e = 1, \kappa = 1.0$ also does, suggesting that some tasks with large domain gaps benefit from more editing steps. Interestingly, $e = 2, \kappa = 0.5$ always slightly outperforms $e = 1, \kappa = 1.0$. This observation indicates that xTED, even with the same computational effort in noising and denoising, can achieve improved performance by dividing the editing process into finer-grained refinement stages.

Table 9: xTED with different editing configurations. Averaged over 5 random seeds.

| Target Data | Source Dynamics | $e = 1, \kappa = 1.0$ | $e = 1, \kappa = 0.5$ | $e = 2, \kappa = 0.5$ |
|-------------|-----------------|------------------------|------------------------|------------------------|
| WK-MR | Gravity | 27.1±10.8 | 23.3±9.1 | **28.3±9.2** |
| | Friction | 25.5±13.6 | 25.9±9.1 | **26.8±8.9** |
| | Thigh Size | 24.1±9.3 | **25.9±9.1** | 24.7±11.0 |

## C.4 XTED ON LARGER AMOUNT OF SOURCE DATA

In this section, we present the results of leveraging a larger amount of source data (200k transitions), i.e. 10 times the target data. We tested policy performance using the TD3+BC algorithm with target data from the WK-MR and HC-MR datasets. As shown in Table 10, xTED demonstrates a clear advantage over the baselines and improves on the results obtained with only 20k source data. These results suggest that xTED is robust and stable in accommodating larger amounts of source data,

highlighting its potential for scaling up model capacity and data usage in future developments of larger-scale cross-embodiment policy learning for robotics.

Table 10: Average normalized scores for MuJoCo tasks on 20k target data and 200k source data.

| Target Data | Source Dynamics | Target | Target+Source | Δ | Target+Edited Source | Δ |
|---|---|---|---|---|---|---|
| HC-MR | Gravity | 26.2±3.5 | 30.9±2.9 | +17.9% | **33.9±2.3** | +29.4% |
| | Friction | 26.2±3.5 | 26.4±3.3 | +0.8% | **28.0±3.2** | +6.9% |
| | Thigh Size | 26.2±3.5 | 23.7±5.4 | -9.5% | **31.0±3.7** | +18.3% |
| WK-MR | Gravity | 16.6±5.9 | 24.5±13.6 | +47.6% | **26.7±14.8** | +60.8% |
| | Friction | 16.6±5.9 | 27.9±10.8 | +68.1% | **36.2±11.7** | +118.1% |
| | Thigh Size | 16.6±5.9 | 21.7±12.1 | +30.7% | **27.6±14.7** | +66.3% |
| | Total | 128.4 | 155.1 | +20.8% | **183.4** | +46.6% |

## C.5 COMPARISONS ON TRANSFORMER AND TEMPORAL U-NET FOR DIFFUSION-BASED DATA EDITING

As shown in Table 11, compared to Temporal U-Net commonly used in diffusion models, our architecture achieved higher overall scores and significantly reduced training time per epoch (1000 training steps). The training time consumptions (seconds) were recorded on a server with $8\times$ A100 GPUs while each model was trained using a single GPU on the server, with no other GPUs active during the process. The detailed architecture of Temporal U-Net can be found in (Ajay et al., 2023a).

Table 11: Comparisons xTED and Temporal U-Net for trajectory editing.

| Target Data | Source Dynamics | **Temporal U-Net** | | **xTED** | |
|---|---|---|---|---|---|
| | | Avg. Score↑ | Clock Time↓ | Avg. Score↑ | Clock Time↓ |
| HC-MR | Gravity | 27.9±3.2 | | **31.3±2.9** | |
| | Friction | 27.7±3.7 | 74.6s | **31.8±3.1** | 46.1s |
| | Thigh Size | 28.8±3.5 | | **33.0±3.0** | |
| WK-MR | Gravity | **30.9±9.6** | | 23.3±9.1 | |
| | Friction | **26.3±12.6** | 93.2s | 25.9±9.1 | 46.5s |
| | Thigh Size | **28.2±9.3** | | 25.9±9.1 | |
| | Total | 169.8 | 167.8s | **171.2** | 92.6s |

## C.6 XTED WITH ADDITIONAL CONSTRAINTS AND CONDITIONS ON DIFFUSION MODELS

We train xTED with constraints of fixed initial transition (Init. Const.) and no constraint of fixed last transition (Last Const.) in trajectories. xTED can also accommodate conditioning strategies (Conditioned xTED). In this section, we ablate on these additional constraints and conditions.

In terms of the implementation of return-conditioned xTED, to generate trajectories that maximize return, we condition the diffusion model on the return of trajectories, such that the training objective in Eq. 5 should be adapted to:

$$\mathcal{L}_\theta = \mathbb{E}_{k\in[1,\cdots,K],\tau_0\sim q(\tau_0),\epsilon\sim\mathcal{N}(\mathbf{0},\mathbf{I})}\|\epsilon - \epsilon_\theta(\tau_k, R(\tau), k)\|^2 \tag{13}$$

where the returns are normalized to ensure $R(\tau) \in [0,1]$. Sampling a high-return trajectory corresponds to setting $R(\tau) \to 1$. In the planning stage, we denoise the blurred trajectories with classifier-free guidance with perturbed noise (Ajay et al., 2023a):

$$\hat{\epsilon} := \epsilon_\theta(\tau_k, \varnothing, k) + \omega\left(\epsilon_\theta(\tau_k, R(\tau), k) - \epsilon_\theta(\tau_k, \varnothing, k)\right) \tag{14}$$

where we assign $R(\tau) = 0.9, \omega = 1.0$.

As shown in Table 12, the use of initial constraint benefits the final performance, while using last constraint has the opposite effect. Notably, incorporating return guidance can further enhance the effectiveness of xTED.

Table 12: xTED with additional constraints and condition guidance. Averaged over 5 random seeds.

| Target Data | Source Dynamics | w/o Init. Const. | w/ Last Const. | xTED | Conditioned xTED |
|---|---|---|---|---|---|
| HC-MR | Gravity | 28.6±4.3 | 21.8±3.3 | **31.3±2.9** | 30.5±3.1 |
| | Friction | 26.3±4.1 | 22.0±3.7 | 31.8±3.1 | **35.1±1.8** |
| | Thigh Size | 29.6±3.4 | 21.3±3.9 | 33.0±3.0 | **33.3±3.0** |
| WK-MR | Gravity | **24.5±7.7** | 18.7±6.9 | 23.3±9.1 | **24.5±9.0** |
| | Friction | 20.6±7.5 | 15.2±6.7 | **25.9±9.1** | 22.3±8.7 |
| | Thigh Size | 18.2±7.3 | 14.5±7.0 | 25.9±9.1 | **31.1±11.5** |
| | Total | 147.8 | 113.5 | 171.2 | **176.8** |

## C.7 xTED with Different Choices of Trajectory Horizons for Diffusion Modeling

In sequential modeling, the sequence horizon is a critical parameter. Modeling sequences that are too long can significantly challenge the model capacity, while sequences that are too short often result in reduced long-horizon consistency. In this section, we compare a shorter horizon of 10 with the horizon of 20 used in our main simulation experiments. As shown in Table 13, selecting a horizon of 20 improves the model performance in the HalfCheetah and Walker2d locomotion tasks.

Table 13: Average normalized scores for MuJoCo tasks on horizon choices for trajectory modeling.

| Target Data | | Source Dynamics | Target+Edited Source (**horizon=20**) | Target+Edited Source (**horizon=10**) |
|---|---|---|---|---|
| Halfcheetah | Med | Gravity | **40.6±2.0** | 40.1±2.4 |
| | | Friction | **41.2±1.6** | 40.6±1.6 |
| | | Thigh Size | **40.7±2.4** | 40.7±2.2 |
| | Med-R | Gravity | **31.3±2.9** | 28.8±3.3 |
| | | Friction | **31.8±3.1** | 29.8±2.8 |
| | | Thigh Size | **33.0±3.0** | 30.2±3.2 |
| | Med-E | Gravity | **43.8±3.6** | 43.6±3.6 |
| | | Friction | **43.2±3.0** | 37.6±3.8 |
| | | Thigh Size | **43.0±3.0** | 40.6±3.5 |
| Walker2d | Med | Gravity | **58.2±11.7** | 53.1±12.6 |
| | | Friction | 54.5±13.7 | **58.9±12.5** |
| | | Thigh Size | **58.9±11.7** | 57.8±12.0 |
| | Med-R | Gravity | **23.3±9.1** | 21.8±7.1 |
| | | Friction | **25.9±9.1** | 25.0±9.1 |
| | | Thigh Size | **25.9±9.1** | 18.9±7.5 |
| | Med-E | Gravity | **82.9±18.1** | 76.0±20.0 |
| | | Friction | **74.0±24.5** | 73.2±16.2 |
| | | Thigh Size | **81.0±21.4** | 75.1±22.9 |
| | | Total | **833.2** | 791.8 |

## C.8 xTED on Different Degrees of Dynamic Gaps

We explore the impact of different degrees of dynamics gap on policy performance aided with xTED. Specifically, we modify the thighs of HalfCheetah and Walker2d to $0.5\times$, $2\times$, and $3\times$ original sizes respectively. The results in Fig. 6 indicate that under various degrees of dynamics gaps, edited source data can augment the original dataset effectively in most of situations, while directly augmenting with unedited source data is sometimes more sensitive to extreme domain gaps in HalfCheetah environments.

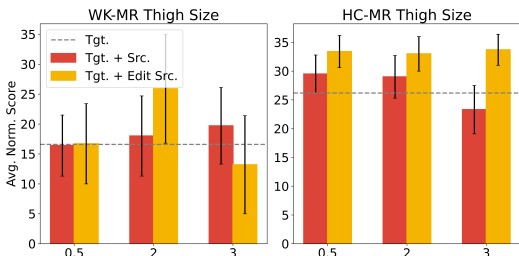

Figure 6: Average normalized returns for different degrees of dynamic gaps (different thigh sizes) on 20k transitions from WK-MR and HC-MR.

### C.9 xTED FOR MULTIPLE SOURCE DOMAINS WITH DIFFERENT GAPS

xTED demonstrates the ability to handle diverse source domains with varying domain gaps using a single pre-trained diffusion model over the target domain, without the need for re-training or fine-tuning. We integrate source datasets exhibiting three distinct domain gaps (20k data with thigh size, gravity, and friction gaps, respectively) into the policy learning process. While unedited source data with intricate domain gaps generally degrade overall policy performance, the edited source data, with narrower gaps, consistently improves policy learning outcomes.

Table 14: Average normalized scores for multi-source MuJoCo experiments.

| Target Data | Tgt | Tgt+Src | Δ | Tgt+Src(Edited) | Δ |
|---|---|---|---|---|---|
| HC-M | 39.5±2.4 | 37.6±3.1 | -4.8% | **39.7±1.9** | +0.5% |
| HC-MR | 26.2±3.5 | 26.6±3.5 | +1.5% | **28.2±3.2** | +7.6% |
| HC-ME | 40.1±2.9 | 35.3±5.0 | -12.0% | **41.3±1.8** | +3.0% |
| WK-M | 45.3±15.9 | 41.8±19.1 | -7.7% | **58.5±9.3** | +29.1% |
| WK-MR | 16.6±5.9 | 19.8±6.2 | +19.3% | **22.7±7.6** | +36.7% |
| WK-ME | 71.0±21.0 | 74.2±15.2 | +4.5% | **81.3±18.0** | +14.5% |
| Total | 238.7 | 235.3 | -1.4% | **271.7** | +13.8% |

