# OpenReview forum: "xTED: Cross-Domain Adaptation via Diffusion-Based Trajectory Editing"
_ICLR.cc/2025/Conference — Submitted to ICLR 2025_

### Official Review · Reviewer_r2FD · 2024-10-28

**Soundness:** 2
**Presentation:** 3
**Contribution:** 2
**Rating:** 5
**Confidence:** 5

**Summary:**

The paper introduces xTED, a cross-domain adaptation framework that applies a diffusion-based approach to edit trajectories from source domains, enabling them to better align with the target domain's dynamics. Through extensive simulations and real-robot experiments, xTED demonstrates enhanced policy learning efficiency, particularly by improving performance.

**Strengths:**

- The paper is well-structured and easy to follow, making the proposed method and experimental results accessible to readers.
- The framework is presented with thorough detail, and a clear understanding of the experimental setup.

**Weaknesses:**

- The effectiveness of the xTED methodology proposed in this paper is questionable, given that there is a considerable overlap in confidence intervals with baselines (Tgt + Src), suggesting similar performance levels. In particular, in Table 1, a significant portion of the results shows comparable performance to the datasets composition approach, with confidence intervals entirely overlapping for all results in the Walk2D environment.
- There is insufficient comparison with SOTA baselines. It seems necessary to include comparisons with methods that integrate cross-domain environments in Offline RL, such as [1, 2]. The studies in [1, 2] include experiments that are particularly similar to those conducted in this paper. Although the problem setups are not entirely identical, cross-domain RL approaches, as explored in [3, 4], are also worth considering for comparison.

[1] Liu, Jinxin, Hongyin Zhang, and Donglin Wang. "Dara: Dynamics-aware reward augmentation in offline reinforcement learning." *arXiv preprint arXiv:2203.06662* (2022).

[2] Liu, Jinxin, et al. "Beyond ood state actions: Supported cross-domain offline reinforcement learning." *Proceedings of the AAAI Conference on Artificial Intelligence*. Vol. 38. No. 12. 2024.

[3] Rao, Kanishka, et al. "Rl-cyclegan: Reinforcement learning aware simulation-to-real." *Proceedings of the IEEE/CVF Conference on Computer Vision and Pattern Recognition*. 2020.

[4] Raychaudhuri, Dripta S., et al. "Cross-domain imitation from observations." *International Conference on Machine Learning*. PMLR, 2021.

**Questions:**

- If the model learns to edit random datasets from the source domain into expert datasets in the target domain, distributional discrepancies could lead to significant issues. Are there any experimental results or discussions addressing this potential problem in the paper?
- In Table 2, Reducing the amount of target data actually leads to improved performance. Why does performance increase with less data usage?

---

### Official Review · Reviewer_gys7 · 2024-10-28

**Soundness:** 1
**Presentation:** 2
**Contribution:** 2
**Rating:** 5
**Confidence:** 4

**Summary:**

The paper addresses cross-domain trajectory adaptation for RL/imitation learning settings, focusing on improving policy learning from data with significant domain gaps, such as those in dynamics, morphology, or visual properties. The authors propose xTED (Cross-Domain Trajectory Editing), a diffusion model-based framework that "edits" source domain trajectories to align with target domain characteristics. The model architecture encodes state, action, and reward sequences separately to maintain dependencies and temporal consistency in trajectories, followed by a denoising process that adapts source trajectories to fit the target domain. The novelty lies in proposing an intuitive sequencing model architecture and introducing regulated noisy source trajectories for training target diffusion models.

The evaluation includes simulation-based experiments in MuJoCo environments (e.g., Walker2d-v2, HalfCheetah-v2) and real-world robotic manipulation tasks. In these experiments, xTED's performance is compared with baselines using success rate, normalized score, and cumulative reward as metrics. Baselines include direct policy learning on target data, using unprocessed source data, and traditional data augmentation techniques.

**Strengths:**

1. According to the extensive experiment results, it seems the following intuitions are valid:
    - separating sequence models with causality interactions for states, actions, and rewards captures their dynamics structures better.
    - injecting noisy source trajectories into the target denoising process as a form of "augmenting diffusion" increases learning performance (Table 1). This could be a good insight into data augmentation for diffusion models.

**Weaknesses:**

**Major points**

1. The main concern is the lack of theoretical investigation of introducing the noisy source trajectories to train the target diffusion model, despite the claim: "this noised initialization theoretically yields favorable results by solving the reverse stochastic differential equation." on Line 276. "Editing" process described in Sec. 4.2. does not form a backward (denoising) process since the noisy source trajectories are not obviously Brownian noise, hence the convergence results [3] do not apply. xTED should be clarified on how it differs from standard diffusion processes and its implications for convergence guarantees. Then, the authors should:
   - provide a rigorous theoretical analysis of xTED or,
   - explicitly state the limitations of applying existing theoretical results [3] on xTED.

2. No assumptions are made about the dimensionality or (domain) structure of source-target state/action trajectories, including states, actions, and rewards. This information is crucial to understanding the scope of cross-domain learning and theoretical understanding of the denoising process with noisy source trajectories. The authors should explicitly state any assumptions on the structure and dimensionality of the source and target domains and discuss how these assumptions impact the applicability and performance of xTED (e.g., how xTED might need to be adapted for domains with different structures or dimensionalities, etc.)

3. Connecting to "imaging editing" is confusing with this paper's formulation. The author should consider revising their framing of the method, by providing a more direct explanation of how they augment the target denoising process with noise-regulated source trajectories. Perhaps a brief explanation to motivate the connection to image editing should be provided.

**Minor points**

4. Sec 2.2 should additionally cite [1, 2] as they formulate diffusion models for smooth trajectories, which are relevant to this paper's claims.
5. There is a typo in Eq. 4. There should be a minus between the $\mu$s.
6. While Sec 4.2 is relatively understandable after carefully reading, it would be much clearer if a small visualization for Sec. 4.2 is presented similarly to Fig. 2. For example, a high-level intuition drawing of the training target diffusion model with noises as perturbed source trajectories.

[1] Carvalho, Joao, et al. "Motion planning diffusion: Learning and planning of robot motions with diffusion models." 2023 IEEE/RSJ International Conference on Intelligent Robots and Systems (IROS). IEEE, 2023.

[2] Luo, Yunhao, et al. "Potential based diffusion motion planning." ICML (2024).

[3] De Bortoli, Valentin. "Convergence of denoising diffusion models under the manifold hypothesis." TMLR (2022).

**Questions:**

1. Please address the major points and minor points above.
2. How does xTED handle/extend to learning between source-target embodiments with different dimensionality and kinematics? (e.g., robot hand with gripper and hand with 3-4 fingers)
   - This question would shed light on fundamental limitations, leading to valuable insights about the method's generalizability and potential future research directions.

---

### Official Review · Reviewer_6aLs · 2024-11-02

**Soundness:** 2
**Presentation:** 2
**Contribution:** 2
**Rating:** 5
**Confidence:** 4

**Summary:**

This paper presents a new method to improve the policy learning on the task where the data collection is laborious or costly. The authors use the diffusion model to transfer the source data which shares primitive skills with the target data, thereby augmenting the dataset for policy learning. During the training phase, the diffusion model acquires prior knowledge from the limited target data. During evaluation, this prior is utilized to filter out task-irrelevant information, retaining only the pertinent skill knowledge from the source trajectory for the target task.

**Strengths:**

i) This paper delicately models the interdependencies among states, actions and rewards within trajectories to facilitate learning in the diffusion model. This detailed modeling significantly contributes to the empirical success of the methodology.

ii) The extensive evaluation results underscore the effectiveness of trajectory editing using the diffusion model.

**Weaknesses:**

i) This model primarily relies on the diffusion model for transferring source data to target data as a form of data augmentation. However, the lack of a theoretical foundation raises concerns about the novelty of this approach, despite the authors' detailed trajectory modeling in the embedding space using neural networks. Therefore, I suggest that the authors conduct a theoretical analysis to elucidate how the diffusion model minimizes the distance between the source data and target data through transformations applied to the source data at the trajectory level. Subsequently, they can theoretically expound on how these minimized distances influence the policy's performance compared to scenarios without data transformations facilitated by the diffusion model.


ii) This work exhibits superiority in scenarios where the source and target domains share similar skill knowledge. My concern is the success of this method may partially stem from the fact that skills inherent in the source data can inspire skill acquisition for the target task. In practical applications, various types of source data are available, and it may not be clear which sources can expedite learning for the target data. It remains uncertain in this study whether the diffusion model-based trajectory editing can effectively handle "bad" samples within the source data for the target task. Therefore, I recommend that the authors test their method on source domains with varying degrees of similarity in skills to the target domain, or assess how the method performs when presented with a combination of pertinent and extraneous source data.


iii) A minor suggestion is to increase the font size in Fig.1 and Fig.2 for improved readability.

**Questions:**

See ii) in the weakness part.

---

### Official Review · Reviewer_HBcs · 2024-11-04

**Soundness:** 3
**Presentation:** 3
**Contribution:** 3
**Rating:** 6
**Confidence:** 3

**Summary:**

This paper proposes a method named "Cross-Domain Trajectory EDiting (xTED)" for adapting data from a source domain into a target domain for better policy learning. The setting considered is that there is some sort of gap between the source and target domain, e.g., dynamics, observations, or agent embodiments, thus the data -- decision trajectories composed of (s, a, s') in a MDP -- from the source domain cannot be directly applied to the target domain, and needs to be first adapted/edited. The proposed method first trains a diffusion model on the target domain data; then, it adds noise to the source domain data, and then apply the trained diffusion model to denoise it to "edit" the noised source trajectory to target trajectory. The paper proposes a new architecture for the diffusion model, which first separately encodes the state, action, and reward via self attention, and then perform cross attention between them, with "Dependency structure modeling." where, e.g., reward only attends to (state, action) pairs. This shows to perform better than traditional architectures such as simply concatenating state, action, reward pairs. Experiments are performed on two simulation tasks, where the source and target domain differs in transition dynamics, and three real-world manipulation tasks, where the source and target domain differs in observation and dynamics, and the proposed method is shown to outperform baselines. Ablation studies are performed to help understand the performance of different components of the method.

**Strengths:**

- The paper is overall clearly written.
- The proposed method is simple, yet seems to be effective.
- The ablation studies are comprehensive and helps to understand the performance gains of the method.

**Weaknesses:**

- One major issue is that lack of strong baselines. There are several baselines on policy/trajectory transfer mentioned in the introduction, e.g., Eysenbach et al, 2021, however, the paper does not compare to any of them. It is mentioned in the intro that these methods employ specific assumptions and thus come with specific constraints, but it would be good to specify, in the context of this paper's experiments, why these methods are not applicable. If they are applicable, then a comparison would greatly strengthen the paper.
- In the abstract, the paper claims "extensive" sim and real experiments. Although this is a subjective measure, but 2 sim envs + 3 pick-and-place tasks for real-world robots may not seem "extensive". I would encourage the authors to test at least on a few more sim envs.
- The proposed method edits the source trajectory by adding noise to the source trajectory, and then use the diffusion model trained on the target data to denoise it. I wonder if this is any theoretic analysis or guarantee on this is a valid operation, i.e., the resulting trajectory will be the target trajectory. In my understanding, the diffusion denoise process needs to start from a Gaussian noise to ensure the correctness of the denoising process. In the paper, however, it starts from a "biased" data which is noise added to source data. A discussion on this would be appreciated.

**Questions:**

Please see the weakness section.

---

### Meta-Review · Area_Chair_ZJ41 · 2024-12-19

**Metareview:**

**summary**

The paper introduces xTED, a diffusion model-based framework for adapting source domain trajectories to align with target domain characteristics, addressing challenges in RL and IL where domain gaps exist (e.g., in dynamics, observations, or agent embodiments). xTED first trains a diffusion model on limited target domain data, then applies noise to source domain trajectories before denoising them to generate target-aligned trajectories. The authors also propose a new architecture, which separately encodes state, action, and reward sequences via self-attention. Extensive experiments on both simulated environments (e.g., MuJoCo tasks with varying dynamics) and real-world robotic manipulation tasks demonstrate that xTED outperforms baselines.

**strengths**

* The paper is clearly written
* Extensive evaluation results

**weaknesses**

* The paper does not compare xTED against sufficiently robust or diverse baselines, limiting the strength of its empirical validation. Also, the overlapping confidence intervals in the evaluation results raise concerns about the statistical significance of the reported improvements.
* The experiments do not explore varying degrees of similarity between source and target domains or scenarios with mixed relevant and extraneous source data, which would provide deeper insights into the method's robustness.

**decision**

Despite its simplicity,  it is difficult to say that the contributions of this work are very significant due to several limitations listed in **weaknesses**. I recommend that the authors address these concerns and consider resubmitting to another venue.

**Additional Comments On Reviewer Discussion:**

Additional experimental results were not enough to address the reviewer's concerns on baselines and statistical significance.

---

### Decision · Program_Chairs · 2025-01-22

Reject